# Efficient Multistage Inference on Tabular Data

Daniel Johnson[1]   Igor L. Markov[2]

[1]Stanford University
[2]Meta

**Abstract**  Many ML applications and products train on medium amounts of input data but get bottlenecked in real-time inference. When implementing ML systems, conventional wisdom favors segregating ML code into services queried by product code via Remote Procedure Call (RPC) APIs. This approach clarifies the overall software architecture and simplifies product code by abstracting away ML internals. However, the separation adds network latency and entails additional CPU overhead. Hence, we simplify inference algorithms and embed them into the product code to reduce network communication. For public datasets and a high-performance real-time platform that deals with tabular data, we show that over half of the inputs are often amenable to such optimization, while the remainder can be handled by the original model. By applying our optimization with AutoML to both training and inference, we reduce inference latency by 1.3x, CPU resources by 30%, and network communication between application front-end and ML back-end by about 50% for a commercial end-to-end ML platform that serves millions of real-time decisions per second. The crucial role of AutoML is in configuring first-stage inference and balancing the two stages.

## 1 Introduction

The recent availability of sophisticated ML tools [29; 1] fueled the development of many new data-driven applications, but considerable efforts are required to engineer robust high-performance ML systems with sufficient throughput to make a practical impact [7; 20; 2; 17; 19; 27]. Other than newer systems designed with ML in mind, a greater variety of existing production systems can be enriched with ML capabilities by modeling the operating environment when closed-form descriptions are not available, helping avoid redundant work and optimizing interactions between system components, also predicting user behaviors and preferences to enhance user experience [25]. As data patterns change, regular retraining, monitoring, and alerts add significant software complexity, but this ML complexity should not overburden product code. To ensure SW development velocity and enable performance optimizations, ML code is often separated into libraries and services — data collection, model training and offline evaluation, real-time inference [26; 17], etc — invoked from product code via RPC APIs.[1] Unlike the well-publicized large language models (e.g., GPT-3[5]) and image-understanding models (e.g., CNN models such as ResNet [18] or attention models [34]), many applications must retrain models before data trends change, often on an hourly or daily basis. With dozens to low thousands of features, these models often train on 100K-10M rows of data. Unlike image pixels, video frames, or audio samples, features in *tabular data* often exhibit different scales and do not correlate [16]. Surprisingly enough, ML competitions with tabular data have been dominated not by deep learning models, but by gradient boosting models [8] such as XGBoost [9], LightGBM [22], and CatBoost [30][2], or ensembles including both deep neural networks and gradient boosted decision trees. Even though deep models can be optimized for tabular data [3] for improved performance, gradient boosted decision trees still outperform on tabular datasets and structured

---

[1]RPC calls within our data center are comfortably within the constraints of real-time inference in the deployed ML platform we work with (see Table 3). Therefore we optimize the mean latency and the overall CPU usage.

[2]The popularity of these packages is affected by training efficiency and support for various hardware accelerators.

data in general [33]. Researchers are still working on improving DNNs for tabular data, and have recently shown great progress [21; 15]. Despite this ongoing progress, DNN models consistently suffer known limitations [16], such as struggling with uninformative features. A recent work [4] investigating the black-box nature of neural networks proved that feed-forward networks with piecewise-linear activation functions can be represented exactly by decision trees. The claim is then extended to arbitrary continuous activation functions via piecewise-linear approximation. Practical aspects aside, this suggests viewing neural networks as a collection of subtrees and subnetworks that are optimized to handle different inputs.

We now focus on the bottleneck of many high-performance production ML systems — real-time inference, which dominates resource usage in industry applications (such as recommendation systems, ranking, ads, caching, UI optimizations, etc.) by 1-2 orders of magnitude compared to training [25]. Such applications often require inference latency below the human cognitive threshold of 300 ms, while minimizing CPU latency helps minimizing total CPU resources (network latency can also contribute to CPU resources, but indirectly via network buffers). As noted earlier, deep learning tends to lose out to XGBoost on structured data and, additionally, batch-processing efficiencies available for DNNs are not helpful for real-time inference. Running on CPUs, inference for XGBoost models can be an order of magnitude faster than for DNNs and more compact in memory, and this shifts the inference bottleneck to RPC API calls issued by product code to ML services. The idea explored in our work is to process at least some inferences quickly with a simple model embedded into product code to bring down mean latency when possible and fall back on RPC APIs when necessary. We develop such *multistage inference* in detail and show that it produces consistently good results for various tabular datasets. To reduce API latency and avoid CPU overhead of network communications, it is important to make the first-stage model dramatically simpler than the second-stage model (accessed via RPC), rather than just instantiate the second-stage model with fewer features as done in [24]. In our environment, product code happens to be written in PHP and the first-stage model embedded into it does not rely on any ML packages.[3] Note that first-stage model *training* does not need to be simple, and here we do use existing high-performance ML packages for this purpose [6]. Another critical aspect of our work is how to determine which inputs are served by which-stage model. multistage inference has been explored in [32] which uses a lightweight classifier on computationally constrained IoT devices to decide where to perform inference, in [28] which tries to be energy-efficient by executing "little" deep neural networks as often as possible while reverting to "big" DNNs when necessary, in [11] which decides between a decision tree and a CNN operating on and embedded device, and in [10] which straddles a tradeoff between accuracy and energy consumption by limiting the size of random forest models on low-power embedded devices. For tabular data used in our work, CNNs would be irrelevant and random-forest models would be inferior to SOTA. Our applications have high accuracy requirements as well as much greater available DRAM and much lower latency than networking with low-power embedded devices can allow. We validate our proposed multistage inference in two ways. First, we show that inference quality is largely preserved across diverse public tabular datasets, and that the decline in ROC AUC and accuracy is minor compared to the large performance gain by handling a large fraction of inputs within the product code. Second, we evaluate actual improvements in a high-performance production system [25] and observe a 1.3x speedup in inference latency and a 30% reduction in CPU resource usage.

In Section 2, we outline the rationale behind our approach, key insights, and three implied tradeoffs. In Section 3, we propose the first-stage model called Logistic Regression with Bins (LRwBins) as well as the approach to allocate the inferences between the stages of the model. In Section 4, we discuss implementation of this multistage system. In Section 5, we evaluate this

---

[3]Inference for the first-stage model can also be implemented in hardware.

approach for public datasets as well as in a commercially-deployed ML platform that performs millions of inferences per second. Conclusions and perspectives are given in Section 6.

## 2 ML Rationale and Tradeoffs

Our proposal provisions for the first-stage model to use simple and fast inference algorithms that can be embedded in product code without significantly increasing complexity. This way, we maximize improvements in latency and CPU usage. There is no reason to simplify training, and if we do, the simple model might significantly underperform the more sophisticated model behind the RPC API calls. This tradeoff between the *sophistication of training and inference* leads us to consider Logistic Regression (LR) as an ML component.[4] Indeed, the formula for LR can be implemented directly in any popular programming language without using ML libraries $\left( h_\theta(x) = \frac{1}{1+e^{-\theta^T x}} \right)$ but bare LR is too limited to serve as the first-stage model.

A second tradeoff is between *ML performance and efficiency of inference*: a small sacrifice in model quality (ROC AUC) may bring large gains in inference efficiency. By training the first-stage model on a subset of the (most important) features of the sophisticated model, we can additionally reduce CPU usage — both in the model itself and during feature fetching, which can also be a CPU bottleneck in practice [25]. To address the performance-efficiency tradeoff, we use a third tradeoff — between *performance and input coverage*. In other words, we limit the first-stage model to only some inputs to keep its performance drop (vs the second-stage model) negligible. The fraction of inputs served by the first-stage model (*coverage*) must be sufficiently large to ensure efficiency gains — in practice, 50% is a reasonable target.

To determine which inferences can be handled by a simpler model, we are motivated by linear approximations to high-dimensional separating hypersurfaces. By breaking our datasets up into subsets of data with similar features and subsequently using a simple model for each subset, we can determine which subsets are suitable for simple models. In these subsets of feature space, it is conceivable that linear approximations to a more complex separating surface could do a good job at separating the data as illustrated in Figure 1. Here, the quadrants with red linear approximations to the blue separating curve are candidates to be handled by a first-stage linear model rather than the slower complex model because they do a good job within their respective quadrants (better approximations can be found by LR).

## 3 LRwBins Algorithm

In this section, we introduce our general method of multistage inference called Logistic Regression with Bins (LRwBins) as well as the method of dividing the data into subsets of similar data. See Algorithm 1 for more details. In practice, each subset of similar data can be constructed with the following procedure. We first use a model-free (such as MRMR [12]) or model-based (such as XGBoost feature importance ranking [9]) approach to determine the relative importance of our features. We split each of the $n$ most important features into $b$ bins dictated by the quantiles of the feature over the normalized training set (lines 2-5 in Algorithm 1). Quantiles are used here because there are features with very different distributions and we generally want to distribute the data equally between the bins to adequately train a linear model in each bin. While quantiles work naturally to break up numerical data, we specifically handle Boolean features by naturally splitting into two bins instead of $b$ bins, and categorical data in a similar manner using a one-hot encoding. The $n$ bins that a datum falls into can be considered an ordered tuple (Figure 2). This ordered tuple determines a "combined bin" which contains all of the data falling into the same ordered tuple, thus creating $b^n$ subsets each consisting of similar data (lines 6-9 in Algorithm 1). Note that the number

---

[4]We have also evaluated SVMs instead of LR, but they did not improve performance of our LR-based solution. Additionally, experiments adding quadratic and nonlinear features to the model did not show improved performance.

**Figure 1**: As a motivating example to using linear approximations of high-dimensional separating hypersurfaces, consider some data consisting of two features $(x_1, x_2)$ and label (represented by either a circle or a diamond). First, by looking at only the data points, we see that the data is not *linearly* separable, but that the *nonlinear* blue curve does a good job. If we arbitrarily break up the data into quadrants by the green line, then we can choose red lines that do a good job of separating the data in each quadrant and can be thought of as linear approximations of the blue curve. This motivates using linear models on subsets of the feature space which is what our approach will do.

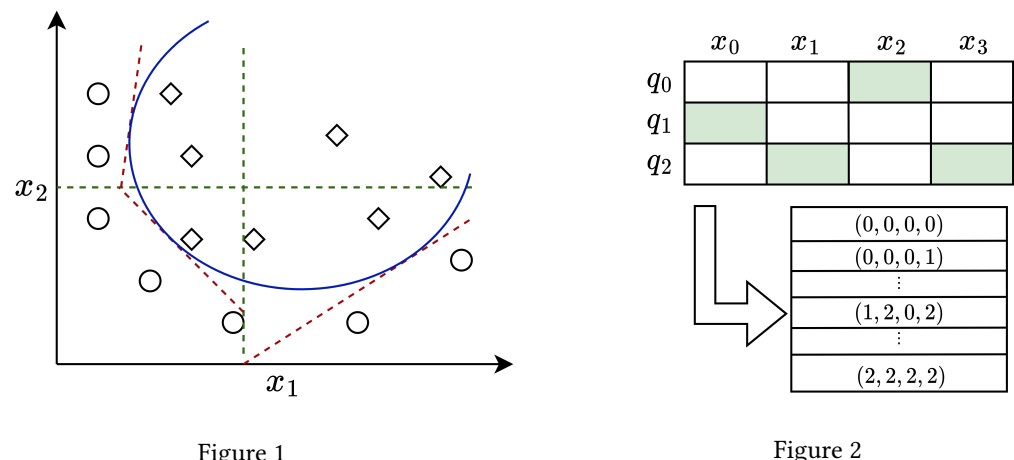

Figure 1

Figure 2

**Figure 2**: This diagram illustrates the mapping of a data point into a combined bin. If each of the $n = 4$ features (represented by $x_i$) are broken into $b = 3$ quantiles (represented by $q_i$), then the ordered pair in which the data point falls into determines the associated combined bin. Each combined bin can store an ML model trained on the data falling into this bin (where enough data is available).

of subsets is the product of the number of bins for each feature, so the total number of subsets may not be $b^n$ when binary features or categorical features are present. In general, since the number of combined bins grows exponentially, both $b$ and $n$ should be kept to reasonably small values to prevent situations where there are many combined bins with very small amounts of data within. In this way, we are essentially building a decision tree that has $b$ branches and depth $n$ and where each split is determined by the quantiles of the data. Continuing with the linear approximation motivation from the previous section, our proposed first-stage model called LRwBins will use an LR classifier within each combined bin (lines 10-13 in Algorithm 1).

For the multistage approach between LRwBins and a secondary, more complex model to work, one must determine how to pick the model to perform the inference. This decision will be split up based on the performance of the models on each combined bin on a validation set of data (see Algorithm 2). Then, during inference, one can simply map the incoming features to a combined bin (similar to line 7 in Algorithm 1), check a stored value to see which model should perform inference (i.e. use the $W_{\text{filtered}}$ lookup table in Algorithm 1), and call the model. To maximize the amount of data using the efficient first-stage model, we proceed as follows. We start by evaluating our desired performance metric (ROC AUC, accuracy, etc) of each model on each combined bin. The combined bins are then sorted by how much the secondary model beats the first model. This means that at the start of the order, we find the combined bins where LRwBins is competitive with or is outperforming the complex model. These bins are ideal for first-stage inference. We combine the first two bins in this order and evaluate the performance metric on the cumulative data. We then add the next bin in this order to the cumulative data, evaluate the performance metric, and repeat until all of the combined bins are being evaluated together. Each evaluation along the way presents an opportunity to split the combined bins between the first-stage and second-stage model. As more and more combined bins are accumulated, the first-stage model handles more inferences

**Algorithm 1** LRwBins($D$ = train data, $V$ = validation data, $b$ = # feat. bins, $n$ = # inference feat.)

1: $F \leftarrow$ RankFeatures($D$)
2: **For** each of the $n$ most important features in $F$ in $D$
3:     **Split** into $b$ bins using quantiles (special handling for Boolean and categorical features)
4: **EndFor**
5: $B \leftarrow$ bin splitting information
6: **For** each datum in $D$
7:     **Determine** combined bin by ordered tuple of $B$ and add datum to combined bin
8: **EndFor**
9: $C \leftarrow$ combined bins
10: **For** each combined bin in $C$
11:     **Train** logistic regression model and save the weights in a lookup table
12: **EndFor**
13: $W_{\text{all}} \leftarrow$ lookup table
14: $S \leftarrow$ TrainSecondaryModel($D$)
15: $W_{\text{filtered}} \leftarrow$ FilterCombinedBins($V, W_{\text{all}}, S$) // see Algorithm 2
16: **return** $W_{\text{filtered}}$

but its ML performance deteriorates. In practice, using the accuracy to determine the combined bin separation gives the best results. Figure 3 explores LR performance per bin and discusses a variant approach to separate data between the first and second stage models.

**Algorithm 2** FilterCombinedBins($V, W, S$)

1: $C_V \leftarrow$ CombinedBins($V, B$)
2: **For** each combined bin in $C_V$
3:     **Evaluate** performance metric for $W$ and $S$
4: **EndFor**
5: **Partition** combined bins based on performance difference between $W$ and $S$
6: **Remove** entries of $W$ corresponding to $S$ partition
7: **return** $W$

Once the combined bins are divided by which model performs inference, the next logical step is to retrain the individual models on the data within their associated combined bins. However, we saw that the first-stage model typically does not see noticeable improvement. The second-stage model also does not improve. This is likely because the gradient-boosted decision tree (GBDT) models often used for binary classification of tabular data generalizes well, so an improvement by training on this subset of the training data would indicate that the original GBDT was not properly capturing all of the data. After separating the data, if we train a new LRwBins model on the data that was not designated for first-stage inference, the new important features on this subset of the data create combined bins which can be evaluated as a second-stage before falling back to the RPC inference. Experiments on production datasets show that this method gave a marginal improvement in the fraction of data handled by the product embedded models meaning that an extra 1 to 3% of the data could be handled by the product embedded model with no model performance loss. For simplicity, we present results with only the first-stage LRwBins model that falls back to the RPC prediction.

## 4 System Implementation

Our practical implementation of multistage inference also includes training and the use of AutoML.

**Figure 3**: To allocate combined bins for inference by first or second-stage models, we evaluate ML performance metrics per bin. Each bar represents a combined bin with the height representing the ROC AUC for that bin, the width representing the number of data rows within each bin, and the color representing the correlation between the global importance of the features (based on the entire dataset) and the local importance of the features (based on the data contained within the bin). The bins are sorted by ROC AUC (or any performance metric such as accuracy) to partition them between first and second stages. A steep dropoff in performance around 50K data rows offers a good separator. Bin-local feature importance shows surprisingly little correlation to global feature importance (for most bins). We explain this by the use of most important features to define combined bins.

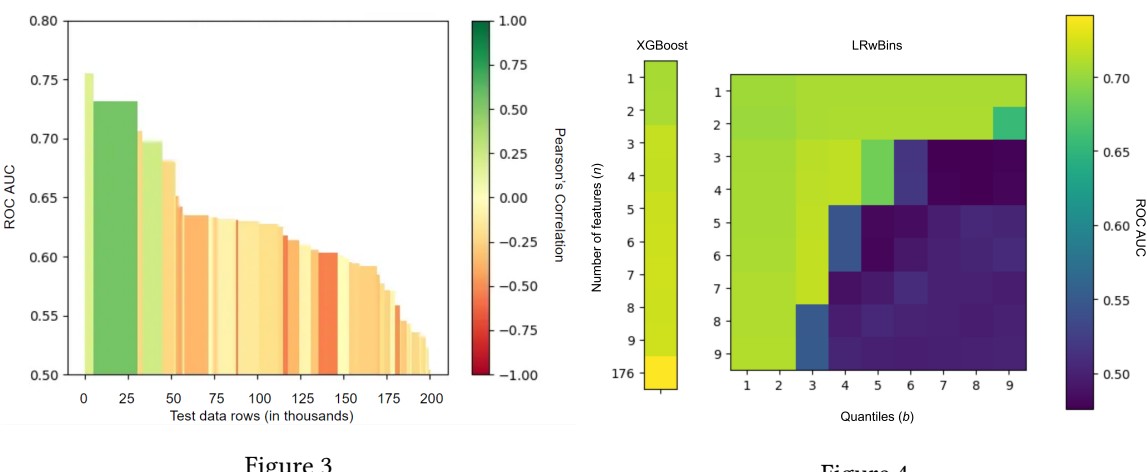

Figure 3

Figure 4

**Figure 4**: AutoML supports automated tuning of parameters ($b$ representing the number of quantiles and $n$ representing the number of most important features to use) on a validation dataset to optimize the shape of the combined bins used by LRwBins. Here we compare the ROC AUC of the LRwBins model for a variety of $n$ and $b$ with the ROC AUC of XGBoost model for a variety of $n$. Notice that we include the ROC AUC for XGBoost using all of the available features (176).

**Training and Inference**. To implement the proposed approach, we train the second-stage model on all of the data to ensure a reliable fallback option for the first-stage model. All training is done with high-performance ML packages while first-stage inference is implemented directly in the product code and reads configuration from a table (rather than loading and running a serialized trained model, as is common in ML platforms today). We checked that our implementations of the first-stage model agree to within machine precision. Compared to XGBoost, LRwBins takes about half the time to train on the same data. To minimize configuration tables for LRwBins, we only store (*i*) quantiles of the $n$ most important features used to determine a combined bin, and (*ii*) LR weights for the combined bins used first-stage inference. An example LRwBins model trained on 1M data rows takes up ~ 0.3KB for the quantiles and ~ 2.3KB for LR weights map with 32-bit floats. Here we present table sizes in RAM, without data compression. During inference, the important features determine the correct combined bin which is used as input to a hash map to get either the LR weights for first-stage inference or a *miss* suggesting to use the second-stage model. If the LR weights are found, the probability is computed using the logistic function and the features.

**The use of ML Automation** is critical to the success of multistage inference, especially to facilitate two of the three tradeoffs described in Section 2, namely the tradeoffs between (2) *ML performance and efficiency of inference*, and (3) *ML performance and input coverage*. We empirically show that input coverage over 50% can be attained with no significant ML performance degradation (Figure 3), but with a marked improvement in efficiency. AutoML is used not only to balance key tradeoffs, but also to configure the first stage of inference. It solves the following tasks: (*i*) determine the shape of combined bins in terms of $b$ (quantiles) and $n$ (important features used) as shown in Figure 4, (*ii*) optimize local models trained on the data in each individual combined bin, and (*iii*) allocate bins

**Figure 5**: Visualizing the features of Case 2 in 2D using [35] clarifies feature selection for inference in the LRwBins model. Each square represents a feature, colors indicate feature types, opacity and geometric proximity to the center reflect feature importance, and integers represent rank by importance.

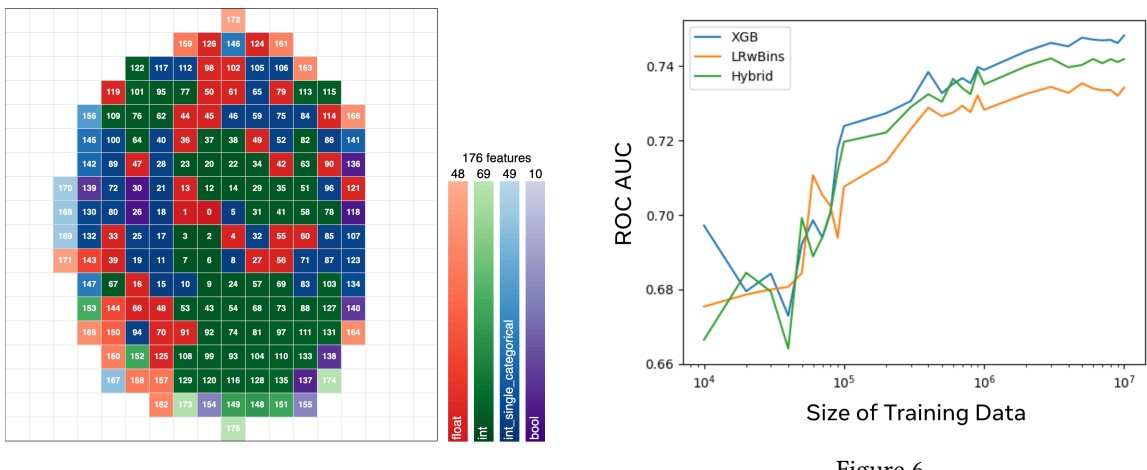

Figure 5

Figure 6

**Figure 6**: Scaling of our multistage approach to 10M data rows in terms of ROC AUC. We compare LRwBins (orange), XGBoost (blue), and the multistage model using each model 50% of the time (green) as we train them on larger subsets of the Case 2 training dataset.

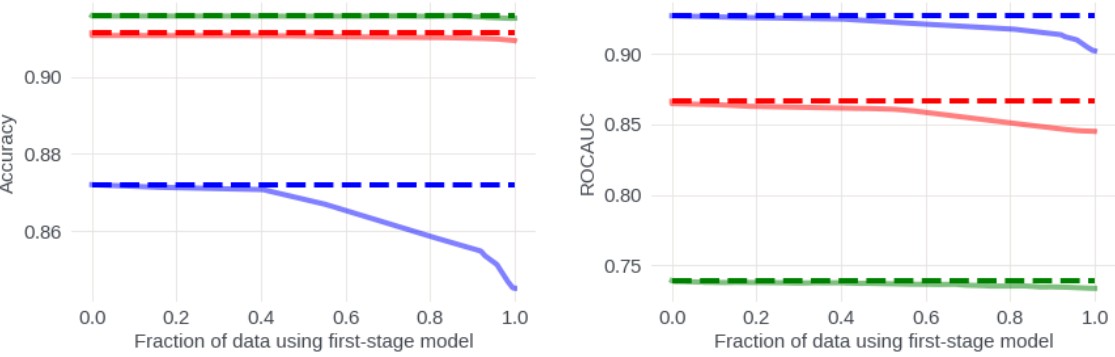

Figure 7: The relative performance of the hybrid models and XGBoost as a function of the percentage of data handled by LRwBins is the central aspect of our argument. We compare these models to multistage inference (solid lines) and XGBoost (dashed line) on several datasets. The very slight decline in ML performance allows a sizable fraction of the data to be handled by LRwBins with minimal loss in performance. Our key insight is that heavy use of the first-stage model entails only a very small ML performance loss. The red is Case 1, the green is Case 2, and the blue is the ACI.

between first- and second-stage models. This is in addition to traditional uses of AutoML to optimize high-performance ML platforms, such as model hyperparameter tuning, feature engineering, and feature selection.

## 5 Empirical Evaluation

We now present results of our multistage inference model that uses LRwBins as a simple first-stage model and XGBoost [9] as a more complex model used via RPC. XGBoost performance is close to that of GBDTs trained as production models. We perform full evaluation on four proprietary datasets from a deployed real-time ML platform. Additional offline evaluation uses the 20+ public

| Dataset | Size | Feats | LR | LRwBins | XGB | LR | LRwBins | XGB |
|---------|------|-------|------|---------|------|------|---------|------|
| | | | ROC AUC | | | Accuracy | | |
| Case 1 | 1000k | 62 | .830 | .845 | .866 | .907 | .909 | .911 |
| Case 2 | 1000k | 176 | .712 | .734 | .739 | .915 | .915 | .916 |
| Case 3 | 59k | 22 | .580 | .615 | .654 | .783 | .785 | .786 |
| Case 4 | 73k | 268 | .565 | .577 | .602 | .900 | .901 | .905 |
| ACI | 33k | 15 | .902 ± .004 | .903 ± .004 | .922 ± .003 | .849 ± .004 | .849 ± .005 | .867 ± .004 |
| Blastchar | 7k | 20 | .839 ± .009 | .839 ± .010 | .839 ± .010 | .800 ± .011 | .800 ± .011 | .798 ± .009 |
| Shrutime | 10k | 11 | .763 ± .010 | .845 ± .006 | .861 ± .008 | .809 ± .006 | .846 ± .006 | .861 ± .004 |
| Patient | 92k | 186 | .860 ± .005 | .872 ± .004 | .899 ± .003 | .926 ± .002 | .926 ± .002 | .932 ± .001 |
| Banknote | 1k | 4 | .879 ± .015 | .938 ± .016 | .989 ± .004 | .801 ± .014 | .838 ± .020 | .947 ± .013 |
| Jasmine | 3k | 144 | .843 ± .017 | .855 ± .017 | .867 ± .012 | .768 ± .017 | .792 ± .016 | .804 ± .015 |
| Higgs | 98k | 32 | .681 ± .004 | .766 ± .004 | .792 ± .003 | .642 ± .004 | .698 ± .003 | .715 ± .003 |

Table 1: A comparison of logistic regression (LR), LRwBins, and XGBoost (a strong baseline model for tabular data) using the ROC AUC and the accuracy as metrics. Cases 1-4 are production use cases on our commercial ML platform. Other datasets are a representative subset of the 20+ public datasets from [33] that we used for evaluation. For each row, we report the mean of 20 random experiments with the standard deviation reported for the public datasets as error.

datasets from [33]. The subset of results reported for public datasets are representative of all of our experiments. We emphasize the improvements in mean latency and CPU usage. We also discuss the limitations of our approach.

## 5.1 ML Performance Benchmarks

Among the public datasets, Adult Census Income (ACI) [23] is based on the 1994 US Census and seeks to predict whether the income of a person is >$50k/year. Blastchar [33] is trying to predict customer retention. Shrutime [33] predicts if a customer closes their bank account or not. Patient [31] dataset looks at the severity of illness. Banknote [13] dataset seeks to determine authenticity based on various factors. Jasmine and Higgs [14] are two datasets chosen from the AutoML benchmark. Cases 1-4 are production use cases that represent a company-internal service, optimize client-server data transfers in a large social network, support user authentication and access to online resources. Figure 5 visualizes the features of Case 2 in 2D using [35] to clarify feature selection for inference in LRwBins. Colors show that the most important features (near the geometric center) include diverse types.

First, we explore standalone performance of LRwBins. By searching over the hyperparameters, we have found that 2-3 quantile bins per feature ($b$) work best and prevent the explosion in the number of combined bins ($b^n$). For larger $b$, many combined bins lack data to train a logistic regression model well. Additionally, about 7 of the most important features used to create the combined bins and 20 features used for inference typically give good results, although these hyperparameters can be tuned for each dataset. In Table 1, we compare logistic regression (LR), LRwBins, and XGBoost across a number of datasets. The LR and LRwBins models use the top $n$ important features determined by hyperparameter tuning while XGBoost always uses all available features. LRwBins outperforms logistic regression and slightly underperforms XGBoost.

As per Section 3, the tradeoff between model performance and inference efficiency comes from deciding which combined bins are handled by which stage of the model. As illustrated in Figure 7 (blue) for the Adult Census Income dataset, increasing the fraction of the data handled by first-stage inference decreases the performance of the ML model in terms of both accuracy and ROC AUC. However, the slight decline in performance of the first 40% of the data provides justification to allow a sizable fraction of the data to be handled by LRwBins with minimal loss in performance. The most important result of this paper is that the initial slope of these lines is so small that the first-stage model can be used on a large fraction of data with only a small ML performance loss. Figure 7 gives representative results on several datasets, but our results on many more datasets

| Dataset | ML Performance Difference (XGBoost Model - Hybrid Model) | | Coverage |
| --- | --- | --- | --- |
| | ROC AUC | Accuracy | |
| Case 1 | 0.003 | 0.000 | 54.2% |
| Case 2 | 0.003 | 0.000 | 49.4% |
| Case 3 | 0.006 | 0.001 | 60.7% |
| Case 4 | 0.010 | 0.002 | 58.4% |
| ACI | 0.002 | 0.001 | 39.1% |
| Blastchar | 0.005 | 0.001 | 24.0% |
| Shrutime | 0.001 | 0.002 | 65.1% |
| Patient | 0.009 | 0.000 | 50.0% |
| Banknote | 0.011 | 0.018 | 60.4 % |
| Jasmine | -0.008 | -0.007 | 53.3 % |
| Higgs | 0.000 | 0.000 | 70.4 % |

Table 2: Analysis of select hybrid models by comparing ML metrics to XGBoost. This hybrid model consists of using LRwBins for a select sensible percentage of the data (i.e. Coverage) and falling back to XGBoost for the remaining data. The percentages are chosen to be as large as possible while allowing for a small tolerance in degradation of ML performance.

(not shown) are similar. Interestingly, a few datasets seemed to show marginal improvement to the XGBoost model at small fractions of data using the first-stage model. As this fraction increased, overall ML performance quickly dropped below the break-even point. Selecting a sensible fraction of data handled by the first-stage model for each of our considered datasets, Table 2 documents small losses in performance metrics. Figure 6 shows that the multistage approach scales well to datasets with millions of data rows and preserves the percentage of data handled by the first stage.

## 5.2 Resource and Latency Improvement

Using multistage inference improves mean latency because the product code directly evaluates first-stage model without latency overhead of ML services. Table 3 shows the total amount of time it takes for a number of first-stage inferences, second-stage inferences via RPC, and multistage inferences. In these experiments, multistage inference is using the first-stage 50% of the time and RPC 50% of the time although this will change based on the dataset as discussed before. We can see that the first-stage inference model is about 5 times faster than the RPC, and the multistage inference is about 1.3 times faster than the RPC. To verify the multistage inference latency, we include a *projected multistage* inference latency time based on the first-stage and RPC latencies. For example, if it takes $t$ time for a RPC prediction (and therefore $.2t$ time for the first-stage prediction), then the multistage prediction should take $.2t$ for half of the inferences. For the other half of the inferences, it will take $.2t$ time to attempt the first-stage prediction and discover that the RPC should be used, and then $t$ time to make the RPC prediction. This all leads to $0.5(0.2t) + 0.5(0.2t + t) = 0.7t$ or 1.4 times speed-up over RPC, close to the empirical 1.3x speed-up. The multistage inference model improved the CPU resource usage as well. While the full model uses all available features, LRwBins fetches only a subset of the most important features (Section 3). In practice, this gave a 1.2x speedup and used 70% of the resources compared to the full model.

## 5.3 Limitations of LRwBins and Unsuccessful Techniques

We found good multistage models for a majority of the 20+ public datasets we experimented with, but a few datasets (less than 10%) benefited little from our approach because they exhibited a steep dropoff in performance with a small fraction of data using the first-stage model. In these cases, the independently-trained second-stage model robustly handles the majority of the inferences.

Additional experiments included using the first $n$ trees trained by XGBoost to similarly bin the data and then train LR models on these bins, but this did not help, and neither did using linear

| Inferences | Average latencies (in milliseconds) for: | | | |
|---|---|---|---|---|
| | 1st-stage Inference | 2nd-stage Inference via RPC | **multistage** | Proj. multistage |
| 10x | 15 | 85 | 82 | 57 |
| 100x | 13 | 65 | 50 | 45 |
| 1000x | 11 | 74 | 57 | 48 |
| 10000x | 8 | 67 | 45 | 42 |

Table 3: Latency for first-stage inferences, inferences that require RPC, as well as measured and projected multistage inferences. The multistage inference latency involves the time for determining which stage should conduct inference, any network latencies between stages, and the time for inference itself. Latency is averaged over inference batches of very different sizes to check for possible measurement overheads. We see that first-stage inference is 5x faster than the second-stage inference, and multistage inference is 1.3x faster than always using second-stage inference. The projected multistage inference latency (based on the first- and second-stage latencies, used 50% of the time) is 1.4x smaller than the second-stage inference, close to our empirical results. We report data for a use case with higher-than-average latency, although other use cases exhibit consistent trends.

SVMs instead of LR in each combined bin. Retraining the networks after splitting the data and adding more stages of inference showed at most negligible improvement in our experiments.

## 6 Conclusion

We introduced and developed an approach to multistage inference that includes a much-simplified first stage that can be embedded into the product code to reduce network communication and lower CPU overhead for a negligible loss in ML performance. The simplified first stage is backed by the full-strength ML model, and AutoML is crucial to configuring the first stage and balancing the two stages. For validation, we used public datasets and company-internal production datasets from a high-performance ML platform that makes millions of inferences per second. In high-performance applications where network latency from RPC APIs is noticeable, the multistage inference approach may be desired to handle up to 50% of the inferences in a quick and efficient manner reducing network communication between the application front-end and ML back-end. The tradeoff between ML performance and inference efficiency can be easily tuned with the LRwBins model which brought a 1.3x drop in latency and 30% drop in CPU usage compared to the RPC prediction.

The use of AutoML to configure the first stage of inference includes optimal choice of hyperparameters when determining the composition of the combined bins. AutoML-based balancing of the two stages determines a separation threshold between the stages, which can vary greatly depending on the use case and the desired tradeoff between (1) ML performance degradation and (2) reduction in computational resources and latency. Optimization of these parameters underlies significant enhancement in resource efficiency and latency, distinguishing it from scenarios where no improvement is achieved. Note that this approach improves average resource-efficiency of inference and thus improves energy-efficiency as well. While additional memory is used, this overhead is small.

Our approach to improve high-performance inference appears compatible with hardware acceleration. We believe that accelerators for LRwBins would be much simpler than DNN-accelerators, use smaller amounts of embedded memory, and likely do well when tree-based ML models outperform DNNs on tabular data. This simplicity comes at the cost of handling only half the inputs without falling back to a heavier model. FPGAs with embedded CPUs appear promising for this application. When dealing with hardware accelerators, AutoML is especially important to tune performance based on specific characteristics of hardware components.

Code for this project is publicly available at: `https://github.com/facebook/lr-with-bins`

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

## 7 Broader Impact Statement

It is difficult to ascribe ethical impact to individual papers that focus more on general-purpose algorithms or approaches meant to optimize performance by reducing resource usage such as this paper. Since this paper focuses on an end goal of maintaining machine learning performance, while reducing the resources used to achieve this goal, we can confidently say that there are no foreseeable potential negative societal impacts that would be brought about by the publication of this paper.

While this paper uses human-derived data for testing and verification purposes, the datasets used do not contain any personally identifiable information (PII) or sensitive personally identifiable information (SPII). We use popular public datasets in order to show performance to familiar data and a few datasets internal to our company representing a company-internal service, optimize client-server data transfers within a large social network, support user authentication, and access to online resources. These datasets also conform to rules and regulations internal to our company.

We have additionally tried to make this paper as accessible as possible to all reviewers and potential audience.

After careful reflection, the authors have determined that this work presents no notable negative impacts to society or the environment.

