# OpenReview forum: "Efficient Multi-stage Inference on Tabular Data"
_automl.cc/AutoML/2023/Conference — AutoML 2023 Workshop_

### Official Review · Reviewer_LbfZ · 2023-04-05

**Potential Impact On The Field Of Automl Rating:** 2
**Technical Quality And Correctness Rating:** 2
**Clarity Rating:** 2
**Actions Required To Increase Overall Recommendation:** 1. Improve clarity
2. Improve empiric…

**Summary Of Contributions:**

The paper presents an approach to speed up inference (on average) in certain deployed applications, through multi-stage inference.  The core proposal is that instead of using remote calls to offload inference to servers, the customer facing or deployed component can perform inference on part of the data.
The approach is meant only for tabular data. The approach achieves this through the proposed Logistic Regression with Bins (LRwBins) algorithm. The authors present a detailed description of the approach, and some example figures to illustrtae the same.  The autyhors also provide a section on implementation. Empirical evaluation is performed with both open source and proiprietary data sets. Additionally, figures are presented to show comparative performance degradation. Furthermore, authors present some additional details on limitations as well as future directions.

**Clarity:**

Clarity is by far the most significant drawback of this paper. The worst thing is the the entire algorithm is verbosely described in Section 3, without any pseudocode. This makes it very difficult for a reader to understand the details in a linear fashion.
Some of the sentences are unnecessarily long(e.g. Line 150).
Figures are not very informative. In Fig 1, the authors explain a familiar concept without the insights & connections to the proposed approach in a clear manner. Similarly, Figures 3 and 4 have long winded captions that do not concisely convey the core ideas. In Figure 7, the legends are missing making it hard to interpret.
The content is not organized well. Tables are few, and tables like Table 1 misses providing the main message.


**Ethics Details (Optional):**

The approach does not seem top present any ethical concerns,

**Overall Review:**

The paper is interesting, but is lacking in several aspects. The narrative clarity is poor and information is not well presented which makes it difficult to understand the details. The technical aspects while sound, authors do not present edge cases or generalization study. The empirical evaluation uses only a few datasets. While some of them are sizable, the applicability of this approach in a broader scenario and different data characteristics is not clear.
On the positive side, the authors provide adequate references. The analysis performed is also good, especially to compare loss of accuracy. The case of RPC causing delays in inference which is a valid problem the authors are proposing a solution for.

**Potential Impact On The Field Of Automl:**

The paper presents a solution to an interesting problem, however the perceived importance or widespread adoption is not clearly evident. Authors do not clearly explain real-world adoption scenarios. While they mention some of the advantages(including things like hardware acceleration on edge devices), it is not clear what is scope and scale where such a problem solution will have an impact.

**Review Confidence:**

3: You are fairly confident in your assessment. It is possible that you did not understand some parts of the submission or that you are unfamiliar with some pieces of related work.

**Review Rating:**

3: Reject: For instance, a paper with technical flaws, weak impact, and/or weak evaluation.

**Review Summary:**

I would recommend the paper not be accepted in the current format.

**Technical Quality And Correctness:**

The basics of the proposed approach appear to be sound. The way the algorithm is presented, is however, makes it difficult to understand the details easily. There are some design choices and ther corresponding effects not clearly explained. Partitioning the data into subregions through feature grouping is explored. Why is dimensionality reduction not considered ?
Line 124: Both b and n should be kept small. But n is data dependent. b i.e quantiles are user provided. It is not clear if b same for all features.
If the cardinality of categorical features is rather high, how does the approach perform ? That is not addressed. LRwBins uses hyperparameter tuning, which is a part of AutoML as the authors noted. But the overall approach does not utilize any other autoML concept significantly.
In terms of novelty, a simpler model to perform inference is very similar to the idea of knowledge distillation. So the application/implementation is novel but the entire idea is not. Some of the points in the conclusion may require further consideration and explanation.

---

> ### Author Response · Authors · 2023-05-02
> **Addressing review**
>
> Dear Reviewer,
>
> Thank you for your valuable feedback on our paper. We appreciate your insights and have made several revisions to address your concerns. Please find our detailed responses below.
>
> Real-world adoption and impact:
> Both the importance and the widespread adoption of the problem we solve have been described in a recent KDD publication not shown during blind review. In particular, our ML platform is deployed for over a hundred real-time industry applications, with millions of inferences per second.
>
> Algorithm clarity and design choices:
> We have added pseudocode to improve the clarity of our algorithm presentation, making it easier for readers to understand the details. We have also fixed the long sentence on Line 150 and updated Figure 1 caption to better connect it to LRwBins. Additionally, we added to the Figure 7 caption clarifying the missing information.
>
> Technical explanations and AutoML integration:
> We have expanded our discussion on using AutoML and its importance as an aspect of the proposed approach. This should provide a more comprehensive understanding of our system and its practical applications.
>
> Additional comments:
> We have made efforts to improve the narrative clarity of our paper by revising sections and adding pseudocode for a better understanding of our proposed algorithm. We acknowledge that presenting edge cases and a generalization study on different data characteristics would be beneficial. However, space constraints make it difficult to include these details in the current version of our paper.
> We plan to investigate these aspects in future work to provide a more comprehensive evaluation of our approach in the arxiv version. Regarding the empirical evaluation, we have added more datasets to demonstrate the applicability of our approach in various scenarios.
>
> Once again, we appreciate your valuable feedback and the opportunity to enhance our work. Please also see our general comment to reviewers on our submission page. Thank you!

---

> > ### Comment · Reviewer_LbfZ · 2023-05-05
> > **Comments on Revisions**
> >
> > A general suggestion is to highlight changes ( preferably in a different color ) to aid comprehensibility in an iterative review process.
> >
> > Positives
> > - Adding pseudocode
> > - Improved clarity to some extent.
> > Negatives
> > - In Alg 1, line 3 -- special handling of categorical variables is not elucidated upon
> > - While the authors have added more datasets, it is not evident to the reader the composition of the features--- since that was one of the issues pointed out earlier that b^n can become large.
> >
> > In summary, the authors have revised the paper to some degree and it is evident. The reviewer's recommendation however does not change owing to aforementioned points.

---

> > > ### Author Response · Authors · 2023-05-05
> > > **Addressing new comment**
> > >
> > > Dear Reviewer,
> > >
> > > Thank you for taking the time to review the edits to our paper and provide feedback. We appreciate your suggestions and will address the concerns you raised.
> > >
> > > Regarding the special handling of categorical variables in Algorithm 1, the special handling is not explicitly elucidated in line 3. However, we do provide a more detailed description of this process on lines 125-127 of the paper.
> > >
> > > We understand your concern about the potentially large size of b^n, particularly for categorical features with high cardinality. In our paper, we have observed that, for the datasets used in our experiments, the important features are not categorical with high cardinality. That is, high-cardinality categorical features, when present, are handled by the second stage. Consequently, we chose not to focus on this less critical aspect. To address this issue, one could either avoid using categorical features with excessively high cardinality when constructing the bins or opt not to use the method altogether in such cases.
> > >
> > > Additionally, we would like to emphasize the robustness of our approach. By incorporating a second stage model, our technique ensures that even when applied to cases with important high-cardinality categorical features, the results do not deteriorate. This aspect further supports the applicability and flexibility of our method.
> > >
> > > Thank you once again for your valuable feedback.

---

### Review · Reproducibility_Reviewer_AHt7 · 2023-04-08

**Completeness Of Code And Dataset Supplement Rating:** 3
**Usability And Ease Of Reproducibility Rating:** 3

**Actions Required To Increase The Reproducibility And Overall Recommendation:**

The readme is 3 lines + 3 section titles long and could benefit from much more detail about the layout of the codebase and high level overviews of each module and full instructions on how to reproduce all results.

Ran into installation issue with current requirements.txt when trying to use latest Python version (3.11) had to fall back to 3.9 to properly install dependencies. Should inform user in README of latest tested supported python version.

The names of the results in Table 1 and 2 do not match the names of the datasets provided in the code base and require the user to debug to find the proper names to reproduce the results.

Plot generation instructions should be provided in the README and the boolean flag to enable should be exposed as a command line argument. Either include an empty "output" directory in the repository or have the evaluate_hybrid_model.py script generate one when needed.

 Submission Checklist 3.e: Hyper-parameters are obscured from the user and require un-pickling to see values for each dataset. No details on how they where selected. The save_hyperparams.py script is provided, but has not mention in the README and no comments or detail explaining what the script does or whether the current version will reproduce the papers results (it does).

The relationship between the lrbins model and the lr model's use of the same model could be better explained with comments.

Having a single command to generate all results (both tables, plots across all 5 datasets with proper names) would be trivial and would make this "push of a button" reproducible.

Overall needs a good code review pass to clean up the code
- All irrelevant code to the implementation and results described in the paper should be removed from the code base (example: the optimize_thresholds functionality in the model/model.py module base class or half of the save_hyperparams.py module)
- All TODOs should be resolved or removed.
- Naming for some methods are confusing (example: get_feature_importances_and_xgb_model does not get anything and rather initializes and trains an entire XGBoost model as a side effect)
- Unnecessary Model abstract class if only used for lrbins implementation (I am guessing it was also going to be used for the lr implementation?)
- nit: DataToBinsMap should use dataclass
- unused variables (example: curpaperxgbrocauc/curpaperxgbacc in evaluate_individual_models.py)

**Completeness Of Code And Dataset Supplement:**

The provided code is able to reproduce the results in both tables 1 and 2 and plot 7. The needed code and data required to recreate table 3 and figure 1-6 are not provided. This data and the platform used to run the tests is the commercial ML platform of the authors.

**Overall Reproducibility Review:**

There are many improvements that could be made to make the code easier to use/read and improve the general quality of the implementations.

The main area for improvement is in the README. Currently there is an over reliance on the user digging through the source code to figure out how to run everything.

Multiple steps and minor code modifications are required to reproduce everything.

**Review Confidence:**

4: You are confident in your assessment, but not absolutely certain. It is unlikely, but not impossible, that you did not understand some parts of the submission or that you are unfamiliar with some pieces of the code or data.

**Review Rating:**

7: Weak Accept, all critical aspects are reproducibile with minor effort, and the remainder are likely reproducible with minor additional effort.

**Review Summary:**

Overall, as far as I can tell, all of the code required to reproduce the results in table 1 and 2 and figure 7, on publicly available dataset, is provided.

Much of the paper uses proprietary datasets from commercial uses cases on the authors ML platform as well as performance benchmarks from that same platform. It is not expected that the reviewer would have access to this platform, but it is unfortunate as the main claim in the paper on CPU and latency savings are based on the aforementioned proprietary performance benchmarks.

With some edits this could be a 9 or 10 rating.

**Summary Of Necessary Code And Dataset Supplement:**

The paper presents a multi-stage inference approach for machine learning on tabular data. The authors propose a method called Logistic Regression with Bins (LRwBins) as the first stage of their multi-stage inference model. The second stage of the model uses a more complex model, in this paper XGBoost, accessed via Remote Procedure Call (RPC) APIs.

The goal of their work is to improve latency and CPU usage in real-time inference on tabular data while maintaining good ML performance.

The authors use AutoML to optimize the shape of combined bins in terms of the number of quantiles and the number of most important features to use. They also use AutoML to optimize local models trained on the data in each individual combined bin and to allocate bins between first- and second-stage models.

The authors evaluate their approach on four proprietary datasets from a deployed real-time ML platform and on 20+ public datasets (5 are provided). They report improvements in mean latency and CPU usage while maintaining good ML performance.


**Usability And Ease Of Reproducibility:**

It was relatively easy to reproduce the results for Table 1 and 2 and figure 7 with the provided code. Limited instructions were provided but with enough review of the code I was able to resolve all issues that I ran into.

- making sure I had the proper version of python installed
- unclear dataset names that do not match the paper results
- no instructions/comments/details on hyper parameter selection
- minor issues generating plots for figure 7 (code/directory modifications needed to run)

---

> ### Author Response · Authors · 2023-04-27
> **Reproducibility review response**
>
> Thank you for the very comprehensive review of our code! We appreciate your feedback and have reacted to it! For each of the points you brought up, we respond with how we changed the code.
>
> Ease of reproducibility
> -----------------------
> Reviewer: It was relatively easy to reproduce the results for Table 1 and 2 and figure 7 with the provided code. Limited instructions were provided but with enough review of the code I was able to resolve all issues that I ran into.
>
> Response: Sorry that you ran into some issues! We have implemented the following changes in response:
> -include a Dockerfile as an additional option to run and explicitly mention the necessary python version needed
> -modified the dataset names to match the paper dataset names
> -included hyperparameters/hyperparametervalues.txt, also included the file print_hyperparams.py and instructions to use it in the README, this prints out the hyperparameters so the user can easily see values; we have included the instructions to generate the hyperparameters as well
> -Plot generation instructions are now in the readme and a command line argument has been added to generate the plots. The output directory has been manually added.
>
> README issues
> -------------
> Reviewer: "The main area for improvement is in the README. Currently there is an over reliance on the user digging through the source code to figure out how to run everything. The readme is 3 lines + 3 section titles long and could benefit from much more detail about the layout of the codebase and high level overviews of each module and full instructions on how to reproduce all results. Ran into installation issue with current requirements.txt when trying to use latest Python version (3.11) had to fall back to 3.9 to properly install dependencies. Should inform user in README of latest tested supported python version."
>
> Response: We have added instructions to specifically reproduce figures/tables and the expected output. We have also added a discussion on the code layout. We regret the installation issue and now specify the python version and include a Dockerfile to run otherwise.
>
> Code changes
> ------------
> Reviewer:
> 1. The relationship between the lrbins model and the lr model's use of the same model could be better explained with comments.
> 2. Having a single command to generate all results (both tables, plots across all 5 datasets with proper names) would be trivial and would make this "push of a button" reproducible.
> 3. All irrelevant code to the implementation and results described in the paper should be removed from the code base (example: the optimize_thresholds functionality in the model/model.py module base class or half of the save_hyperparams.py module)
> 4. All TODOs should be resolved or removed.
> 5. Naming for some methods are confusing (example: get_feature_importances_and_xgb_model does not get anything and rather initializes and trains an entire XGBoost model as a side effect)
> 6. Unnecessary Model abstract class if only used for lrbins implementation (I am guessing it was also going to be used for the lr implementation?)
> 7. nit: DataToBinsMap should use dataclass
> 8. unused variables (example: curpaperxgbrocauc/curpaperxgbacc in evaluate_individual_models.py)
>
> Response:
> 1. Good point! Comments have been added to the train_model class.
> 2. We have added a single command to generate all of the results: ./all.sh
> 3. Irrelevant code has been removed. This includes any model saving code we had that would be used to evaluate the model on the product code side.
> 4. The TODOs have be resolved.
> 5. This has been addressed.
> 6. Yes, this is a great point. We actually had several different models that used the Model class, but we left these out of the paper because they did not perform as well. We have therefore gotten rid of the Model class and everything is in the LRBinsModel class now.
> 7. Fixed
> 8. Fixed
>
> Missing plots
> -------------
> Reviewer: The needed code and data required to recreate table 3 and figure 1-6 are not provided. This data and the platform used to run the tests is the commercial ML platform of the authors. The other could include the code used to generate table 3 and figures 1-6, even tho the reviewers would not have access to the specific API, to at least demonstrate to the reviewers how the results where generated. Much of the paper uses proprietary datasets from commercial uses cases on the authors ML platform as well as performance benchmarks from that same platform.
>
> Response: Unfortunately, we do not have permission to include the data necessary to recreate these results. Additionally, we think that it is a great idea to include the code used to generate things like table 3, but we would need to get permission to include this additional code which is unlikely to be approved until after the rebuttal deadline. We only have permission to open-source and share the code that we have already provided.
>
> Thank you for your help and your feedback provided in your review!

---

### Official Review · Reviewer_BtQb · 2023-04-10

**Potential Impact On The Field Of Automl Rating:** 2
**Technical Quality And Correctness Rating:** 3
**Clarity Rating:** 3
**Actions Required To Increase Overall Recommendation:** NA

**Summary Of Contributions:**

The paper discusses the challenges in developing robust high-performance machine learning (ML) systems and proposes multi-stage inference as a solution to the bottleneck of real-time inference. The authors argue that decision tree models, such as XGBoost, outperform deep learning models on structured data, and that a simple first-stage model embedded in the product code can reduce latency and CPU resource usage. The authors validate their proposed multi-stage inference on diverse public tabular datasets and in a high-performance production system, showing minor declines in accuracy and ROC AUC but significant latency and CPU resource consumption gains.

**Clarity:**

The work is very well written and is very easy to read. However, it has some details that could improve clarity.

The authors seem to relegate the AutoML task to hyperparameter tuning, when in my opinion, the work contributes to the automation of multi-stage inference systems. Clarifying the degree of contribution or the current status of that automation could be something interesting.

Table 1 shows a comparison of logistic regression (LR), LRwBins, and XGBoost, but in table 2 we see the difference between xgboost and hybrid model which is confusing. Is LRwBins the hybrid model? If it is something else, shouldn't it be in table1?

Table 1 leaves the margins. Perhaps you could make the columns smaller by using scientific notation in dataset "size" columns, using .01 instead of 0.01, or using some other shortcut.

More information about the experimental setup would further clarify the contributions of the work.



**Overall Review:**

The paper offers several strengths:

-The paper describes a well-defined problem of creating multi-stage inference systems and proposes a practical solution to automate the process using AutoML.
-The proposed solution appears to be logically consistent with the stated problem and the experiments conducted to evaluate the approach are well-designed, producing results that support the authors' claims.
-The work is very well written and is easy to read.
-The multi-stage method proposed in the paper is a highlight, as it reduces the number of calls to the model's API, saving time and resources.

The authors provide a clear description of the infrastructure experiment setting, which is important for understanding the results and contributions of the work.

There are some concerns with the paper, including:

-As noted in the Clarity section, comparison between XGBoost and a hybrid model is confusing, and it is unclear if the hybrid model is the same as LRwBins or something different. Clarifying this is important to relate and improve traceability between tables 1 and 2.
-More information about the experimental setup could further clarify the contributions of the work. Specifically, information about the infrastructure for the remote model and product code.


**Potential Impact On The Field Of Automl:**

The paper describes the problem of separating data into bins to create multi-stage inference systems, where determining the number of bins and features to use is very important for its success. AutoML is an important tool for accomplishing this task in practice, since allows to automate the process of find good configurations of bins and features. This contributes to the automation of multi-stage inference systems development.

**Review Confidence:**

4: You are confident in your assessment, but not absolutely certain. It is unlikely, but not impossible, that you did not understand some parts of the submission or that you are unfamiliar with some pieces of related work.

**Review Rating:**

8: Accept: Technically sound paper with major impact and strong evaluation, with perhaps some minor flaws.

**Review Summary:**

The paper propose a practical solution to automate the process of creating multi-stage inference systems for real-time inference problem. The experiments conducted to evaluate the proposed approach are well-designed, and the results support the authors' claims. The paper is well-written and easy to read. While there are some minor details that could be improved for clarity, overall, the work presents a valuable contribution to the field of AutoML and Multi-Stage inference.

**Technical Quality And Correctness:**

The authors provide a clear description of the problem they are trying to solve, and their proposed solution appears to be logically consistent with the stated problem. The experiments conducted to evaluate the proposed approach are well-designed and the results appear to support the authors' claims.

One of the highlights of the multi-stage method is that it allows us to save time by reducing the number of calls to the model's API. This time is affected by many factors, such as the infrastructure where the model is located and the network load. Some explanation about the infrastructure experiment setting could be important. Is the network time constant in these experiments? Do the machines for the remote model and product code have the same capabilities?

---

> ### Author Response · Authors · 2023-05-01
> **Addressing comments**
>
> Dear Reviewer,
>
> Thank you for the insightful comments that help improve the manuscript.
> We have clarified the hybrid model confusion in the caption of Table 1 and have also modified Table 1 according to your comments.
> Indeed, our work's contribution to AutoML is beyond hyperparameter tuning, with the main focus on configuring multistage inference to ensure proper tradeoff between efficiency and coverage.  A key empirical result is that the first stage can typically handle over 50% of inferences, and this does not require hand tuning.
>
> We appreciate your valuable feedback and the opportunity to enhance our work. Please also see our general comment to reviewers on our submission page. Thank you!

---

### Official Review · Reviewer_UffP · 2023-04-13

**Potential Impact On The Field Of Automl Rating:** 2
**Technical Quality And Correctness Rating:** 3
**Clarity Rating:** 3

**Summary Of Contributions:**

In many machine learning applications, real-time inference can still present a bottleneck. Hence, it is common to separate ML code into services that are queried by product code using APIs. While this approach simplifies product code by abstracting away ML internals and clarifies software architecture, it also introduces network latency and additional CPU overhead.

In this paper, the authors propose simplifying inference algorithms and embedding them directly into the product code, thereby reducing network communication. Their approach can optimize over half of the inputs for public datasets and a high-performance real-time platform that deals with tabular data. By using AutoML for both training and inference, they achieved a reduction in inference latency, CPU resources, and network communication between the application front-end and the ML back-end.


**Actions Required To Increase Overall Recommendation:**

Better highlight AutoML part.
Try to better highlight the contribution within the global system.

**Clarity:**

The technical aspects of the proposed approach are presented in a clear manner, but the section on the algorithm could benefit from more detail. The authors provide a comprehensive description of their experiments, which are performed on large-scale benchmark applications.

**Overall Review:**

Overall Review

Pro :

The empirical evaluation of the proposed method is based on experiments conducted on both public and company-internal datasets from a high-performance ML platform that serves millions of real-time decisions per second.

The evaluation shows that the proposed method, LRwBin, achieves competitive performance. In particular, the standalone performance of LRwBin demonstrates the efficiency of the approach.

Code for this project is publicly available

Cons

Although the proposed system is mainly based on previously existing techniques, it still makes a valuable contribution to the field by combining these techniques in a novel way.

Some parts of the paper lack technical explanations and details, which could make it difficult for readers to fully understand the system. Moreover, while the AutoML part is mentioned, it could have been better highlighted as an important aspect of the proposed approach.

Additionally, the paper discusses the tradeoff between ML performance and efficiency of inference, but this aspect could have been given more attention and elaboration, especially since it is an important consideration for practical applications.


**Potential Impact On The Field Of Automl:**

The paper includes a possible application of AutoML techniques to optimize a high-performance ML platform in multiple ways, such as model hyperparameter tuning, feature engineering, and feature selection. The results demonstrate that this approach can significantly improve the overall performance of the platform, resulting in better accuracy, faster inference times, and reduced resource consumption.

**Review Confidence:**

3: You are fairly confident in your assessment. It is possible that you did not understand some parts of the submission or that you are unfamiliar with some pieces of related work.

**Review Rating:**

4: Weak Reject: For instance, a paper with minor technical flaws, limited impact, and/or weak evaluation.

**Review Summary:**

The paper proposes a multi-stage inference approach that includes a simplified first stage that can be embedded into product code to reduce network communication and lower CPU overhead with negligible loss in ML performance. The proposed approach relies on AutoML for the optimal choice of hyperparameters, the composition of combined bins, and choosing the separation threshold between stages of inference. The approach improves the average resource efficiency of inference, which also improves energy efficiency, and is compatible with hardware acceleration. This work could lead to significant enhancements in resource efficiency and latency.

This is a technical paper that presents a system together with an extensive empirical evaluation. The outcomes may be important in terms of required resources and are relevant for handling real-time inference in large applications context.

The AutoML contribution is limited since AutoML is only used as a tool in the system. Moreover, this part lacks details.


**Technical Quality And Correctness:**

The proposed approach involves a combination of several technical steps, and although some details are not included due to space limitations, the key aspects are well-explained. The experiments conducted are extensive and performed on large application benchmarks, which strengthens the argument for the effectiveness of the proposed approach. The results presented provide convincing evidence that the approach can significantly improve the performance of the ML platform, with improvements observed in terms of accuracy, inference time, and resource consumption.

---

> ### Author Response · Authors · 2023-05-02
> **Addressing review**
>
> Dear Reviewer,
>
> Thank you for your valuable feedback on our paper. We appreciate your insights and have made several revisions to address your concerns. Please find our detailed responses below.
>
> Highlight impact:
> We note that real-time inference dominates resource usage in industry applications by 1-2 orders of magnitude compared to training.
> In a major difference from academic ML research, industrial applications of ML spend much greater resources on inference, to support products. This can be illustrated by ChatGPT, which reportedly costs \$700M/day to run, but only tens of millions USD total to train. The cost of inference is determined by CPU/GPU resources, which are closely related to mean latency. Whether the mean CPU latency is in seconds or milliseconds, it is multiplied by the large number of queries served to obtain total resources. Additionally, user-observed latency includes network latency and that of database lookups. Many online applications require latency below the cognitive threshold of 300ms, to appear instantaneous.
>
> Algorithm detail and clarity:
> We have added pseudocode and more comments to provide a clearer and more detailed presentation of the algorithm, making it easier for readers to understand the system.
>
> Technical explanations and AutoML integration:
> We have expanded our discussion on using AutoML and its importance as an aspect of the proposed approach. This should provide a more comprehensive understanding of our system and its practical applications.
>
> Tradeoff between ML performance and efficiency of inference:
> While we would have liked to provide more attention and elaboration on this tradeoff, we are limited by the space constraints of the paper. We believe our current discussion provides a reasonable balance between exploring this tradeoff and addressing other important aspects of our work.
>
> Once again, we appreciate your valuable feedback and the opportunity to enhance our work. Please also see our general comment to reviewers on our submission page. Thanks!

---

### Official Review · Reviewer_DuT1 · 2023-04-20

**Potential Impact On The Field Of Automl Rating:** 1
**Technical Quality And Correctness Rating:** 3
**Clarity Rating:** 3
**Actions Required To Increase Overall Recommendation:** Addressing the questions and correcti…

**Summary Of Contributions:**

- The authors propose a solution that offers a speed-up during inference time by using multi-stage inference, switching between a simpler model and a more complex model such as XGBoost based on a certain fraction of the total points. The authors use LR by considering the application at only a subspace at a time, where, only the most accurate subspaces are combined for the first-stage inference.

**Clarity:**

- I could not easily find what the colors stand for in Figure 7 as the description of the figure does not include the datasets (except for blue belonging to adult census in Line 215.)
- Table 1 is breaking the template.
- Line 65 and following, I would personally not use in [reference] and I would stay consistent with the way the references are done in the rest of the paper.

**Overall Review:**

**Questions:**

- Line 123: Creating $b^n$ subsets

    If I understand correctly, for each numerical feature we have b bins, for a binary categorical feature we have 2 bins, while for categorical features we have n_unique_cat bins, so $b^n$ is not the correct term, since for a categorical feature we can have either fewer or more bins, making the term neither a minimum, neither a maximum constraint.

- Could the authors describe the number of features and bins that were discovered for the considered datasets in Table 1?

- Is there any reason for why standard deviations were not provided for Cases 1-4 in Table 1?

- "Other datasets are a representative subset of the 20+ public datasets from [31] that we used for evaluation."

    What was the criteria for filtering the datasets and only including a subset?

- I would like the authors to add a few more datasets to their experiment. In particular, I would request 3 interesting public datasets from the AutoML Benchmark[1], KDDCup09_appetency, jasmine, Amazon_employee_access, and Higgs from [31] (binary classification problems that fit the experimental framework of the work).


[1] Gijsbers, P., LeDell, E., Thomas, J., Poirier, S., Bischl, B., & Vanschoren, J. (2019). An open source AutoML benchmark. arXiv preprint arXiv:1907.00909.

[2] The work.

**Corrections:**

**Line 38: decision trees still outperform on tabular datasets**

I would correct decision trees to gradient boosted decision trees since by themselves decision trees are weak learners.

**Line 35: ML competitions with tabular data have been dominated not by deep learning models, but by gradient boosting models**

 I believe ML competitions quite a few times include an ensembling of DNNs and GBDTs for the top solutions.

**Line 47, Deep learning tends to lose out to XGBoost on structured data and this trend is stronger when training data is limited in size.**

The related work section seems to be lacking, since there have been several works that argue the opposite, that deep learning outperforms tabular data and the works do not mention any trends that deep learning does not perform well when training data is limited [1][2]. From personal experience, I have not observed the trend either.

Kadra, A., Lindauer, M., Hutter, F., & Grabocka, J. (2021). Well-tuned simple nets excel on tabular datasets. Advances in neural information processing systems, 34, 23928-23941.

Gorishniy, Y., Rubachev, I., Khrulkov, V., & Babenko, A. (2021). Revisiting deep learning models for tabular data. Advances in Neural Information Processing Systems, 34, 18932-18943.

**Potential Impact On The Field Of Automl:**

-  I believe the framework proposed by the authors is too problem specific and the speed-up improvements are calculated on a millisecond basis. This could potentially have an impact on an organization that spends a very large fraction of time doing predictions on a set of fixed problems which can decrease the costs of the organization as well as have positive impacts on the environment. The speed-up comes with a trade-off in performance which could make the acceptance into mission-sensitive systems less acceptable.

**Review Confidence:**

4: You are confident in your assessment, but not absolutely certain. It is unlikely, but not impossible, that you did not understand some parts of the submission or that you are unfamiliar with some pieces of related work.

**Review Rating:**

4: Weak Reject: For instance, a paper with minor technical flaws, limited impact, and/or weak evaluation.

**Review Summary:**

Pros:

- The paper proposes an interesting solution on how to handle multi-stage inference and provides a motivating example (Figure 1) on the usage of LR by putting the instances into diverse bins.
- The paper is clear.

Cons:

- The method proposed by the authors affects a specific niche, that is an organization that needs to perform a large number of predictions continuously on an already trained model. I do not believe the proposed method affects the majority of researchers in the field of AutoML, since the time taken even without using the multi-stage inference is in the range of milliseconds. Additionally, the latency reduction comes with a performance deterioration. Moreover, preparing the method has a significant overhead on the practitioner since one needs to detect the feature importances (the authors also suggest XGBoost), then organize the data into bins, run LRwithBins and detect the correct hyperparameters based on the hyperparameter n for the number of important features and b number of bins via HPO. Lastly, the practitioner would need to detect the threshold that is also dataset specific for where the performance degrades from using LRBins and XGBoost.

- There is a larger space overhead that would be required for every dataset depending on the number of features and number of bins.

**Technical Quality And Correctness:**

- The claims mentioned in the paper are sound.
- Verifying the correctness of the claims is difficult since one would need access to the infrastructure that the authors use or to a similar infrastructure, as well as to proprietary datasets.
- Limited evaluations of public datasets, limited evaluation to binary classification tasks only.

---

> ### Author Response · Authors · 2023-05-02
> **Addressing review**
>
> Dear Reviewer,
>
> Thank you for your thoughtful review and valuable feedback on our paper. We have carefully addressed your comments and made the necessary changes to improve our manuscript. Please find our detailed responses below.
>
> Potential Impact On The Field Of AutoML:
> We appreciate your insights on the potential impact of our work. We acknowledge that our method solves a specific practical challenge, but it is an important one. Industrial applications of ML often have significant resource demands for inference to support products, and our method aims to optimize the tradeoffs between performance, speed and resource usage.
>
> Figure 7 colors:
> Thank you for pointing this out. We have updated the caption to include the information about the colors and datasets in Figure 7.
>
> Table 1 formatting:
> We have fixed the formatting issue with Table 1 as you suggested.
>
> Referencing style:
> We have carefully considered your suggestion, but have decided to maintain our current referencing style for consistency throughout the paper.
>
> Line 123: Creating subsets:
> You are correct in your understanding of the binning process. We have revised the terminology in the manuscript to more accurately reflect the constraints of our binning approach.
>
> Features and bins in Table 1:
> Due to page limitations, we have not included these details in the paper. However, we have provided this information in the hyperparameters file of our code supplement, as you suggested.
>
> Dataset selection criteria:
> We chose a representative subset of the datasets showing a few that worked well with our approach and a few that did not work as well.
>
> Additional datasets:
> Thank you for suggesting additional datasets for our experiment. We have included 2 of the suggested datasets and cited the AutoML benchmark paper. Unfortunately, we were unable to include the other two datasets due to technical limitations.
>
> Line 38, 35, 47 corrections:
> We appreciate your suggestions and have made the necessary corrections in our manuscript.
> We have expanded our related work section to include the papers you mentioned and to better reflect the current state of research in the field.
>
> Method's applicability and overhead:
> In a major difference from academic ML research, industrial applications of ML spend much greater resources on inference, to support products. This can be illustrated by ChatGPT, which reportedly costs \$700M/day to run, but only tens of millions USD total to train. The cost of inference is determined by CPU/GPU resources, which are closely related to mean latency. Whether the mean CPU latency is in seconds or milliseconds, it is multiplied by the large number of queries served to obtain total resources. Additionally, user-observed latency includes network latency and that of database lookups. Many online applications require latency below the cognitive threshold of 300ms, to appear instantaneous. We also clarifiy that the overhead that would be required for every dataset depending on the number of features and number of bins is small in general.
>
> We hope that our revisions have addressed your concerns and improved the quality of our paper. Please also see our general comment to reviewers on our submission page. Once again, we appreciate your valuable feedback and the opportunity to enhance our work.

---

### Author Response · Authors · 2023-05-02
**General comment to reviewers and editors regarding reviews**

Dear Reviewers,

Thank you for the time and effort you have invested in evaluating our work. We have carefully addressed the comments provided and improved our revised manuscript as a result.

Notably, most reviewers found our paper to be clear ("The authors provide a clear description of the problem they are trying to solve, and their proposed solution appears to be logically consistent with the stated problem", "The work is very well written and is very easy to read. However, it has some details that could improve clarity", "The technical aspects of the proposed approach are presented in a clear manner, but the section on the algorithm could benefit from more detail").
We further enhanced the clarity based on constructive feedback received:
- We added pseudocode, making it easier for readers to understand the details and follow the proposed method.
- We provided requested details.
- We fixed a variety of small glitches and omissions noted by reviewers, and improved formatting as requested.
- We added two new datasets in the empirical evaluation.
- Admittedly, we may have understated the importance of AutoML in our original manuscript. Therefore, our revised manuscript articulates the crucial role of AutoML in the success of multistage inference --- to facilitate key tradeoffs between the two stages of inference and to configure the first stage.
- We are particularly grateful to the reviewer who provided comments on our code, and we addressed those as well, making the code easier to understand and reuse.

We also addressed several misconceptions raised by the reviewers.
- Optimizing inference for real-time ML systems is not a niche area but rather a mainstream concern in the industry, where efficient ML inference impacts resource consumption and affects business profitability. This is critical in a growing number of real-time ML deployments --- within Web search, ad networks, many applications in social networks, various recommendation systems, etc.
- Optimizing mean inference latency in terms of (tens and hundreds of) milliseconds is not a limitation, but rather a typical industry context. Optimizing mean CPU latency (without increasing the parallelism) helps reduce overall CPU resources.
- The widespread impact of our proposed solution in real-time industry applications should be clearer when we disclose our affiliation and prior publications in the published paper (but we cannot do this during blind review). Our solution is deployed in a real-time industry ML platform that performs millions inferences per second for a variety of applications with billions (with a B) users worldwide.

We trust that our revisions address the reviewers' concerns and enhance the quality of the manuscript, making it suitable for publication.

Thank you once again for your valuable feedback and the opportunity to improve our work.

The Authors